

# Phylogenetics and population genetics of *Plotosus canius* (Siluriformes: Plotosidae) from Malaysian coastal waters

Nima Khalili Samani, Yuzine Esa, S.M. Nurul Amin and Natrah Fatin Mohd Ikhsan

Department of Aquaculture, Faculty of Agriculture, Universiti Putra Malaysia, Serdang, Selangor, Malaysia

## ABSTRACT

*Plotosus canius* (Hamilton, 1822) is a significant marine species in Malaysia from nutritional and commercial perspectives. Despite numerous fundamental research on biological characteristics of *P. canius*, there are various concerns on the level of population differentiation, genomic structure, and the level of genetic variability among their populations due to deficiency of genetic-based studies. Deficiency on basic contexts such as stock identification, phylogenetic relationship and population genetic structure would negatively impact their sustainable conservation. Hence, this study was conducted to characterize the genetic structure of *P. canius* for the first time through the application of mitochondrial Cytochrome Oxidase I (COI) gene, cross amplification of *Tandanus tandanus* microsatellites, and a total of 117 collected specimens across five selected populations of Malaysia. The experimental results of the mitochondrial analysis revealed that the haplotype diversity and nucleotide diversity varied from 0.395–0.771 and 0.033–0.65 respectively. Moreover, the statistical analysis of microsatellites addressed a considerable heterozygote insufficiency in all populations, with average observed heterozygosity ($H_o$) value of 0.2168, which was lower than the standard heterozygosity in marine populations ($H_o = 0.79$). This alongside the high *Fis* values estimation, high pairwise differentiation among populations and low within population variations are supposed to be associated with small sample size, and inbreeding system. Besides, the significant finding of this study was the sharing of common haplotype KR086940, which reflects a historical genetic connectivity between Peninsular Malaysia and Borneo populations due to the geological history of Southeast Asia during Pleistocene era. Demographic analyses showed that all populations were in an equilibrium state with no significant evidence of population expansion. To put it briefly, the current study has managed to provide an initial genomic database toward understanding of the genetic characterization, phylogenetic, molecular diversification and population structure in *P. canius*, and should be necessary highlighted for appropriate management and conservation of species. Further studies must be carried out involving more geographical and sampling sites, larger population size per site, and utilization of species specific microsatellites loci.

Corresponding author
Nima Khalili Samani,
nima.khalili.samani@gmail.com

# INTRODUCTION

*Plotosus canius* (Hamilton, 1822) that is known as grey eel-catfish, black eel-tail catfish, canine catfish or Indian catfish (*Khan et al., 2002*; *Riede, 2004*; *Usman et al., 2013*; *Prithiviraj, 2014*), were diagnosed as a member of genus *Plotosus*, family Plotosidae (*Froese & Pauly, 2015*). They are being mainly distributed in estuaries, freshwater rivers, lagoons, and shallow waters of Australia and Southeast Asia (*Carpenter, 1999*; *Prithiviraj & Annadurai, 2012*). The species is an amphidromous and demersal bony fish that can live in marine, brackish and freshwater habitats. Their relocation in about 100 km range were described as cyclical and frequent horizontal movement on which could not be categorized as breeding migration (*Riede, 2004*). However, the species might be recently endured genetic destruction mostly due to overexploitation similar to other fish species (*Pauly et al., 2002*; *Collette et al., 2011*; *Usman et al., 2013*), their population structure could be considered as the reliable indicator in detection of sustainable and healthy marine environments (*Thomsen et al., 2012*; *Bourlat et al., 2013*).

Population structure is the direct consequence of biogeography (*Leffler et al., 2012*), which provides invaluable statistics on patterns of species dynamics, colonization, and isolation (*Costello et al., 2003*). As species become accustomed to new habitats, the effective size of population extends through its dispersal, resulting in intensification of genetic variation (*Charlesworth & Willis, 2009*). However, deterioration of environmental equations alongside with ecological fluctuations such as recent re-treatment of Pleistocene era have changed species extensive genetic patterns (*Krishnamurthy & Francis, 2012*). Adding to complication, the accuracy of associated conservation strategies can be successively restrained by deficiency of reliable knowledge on biodiversity, conservation resolution, and extent of biological destruction among taxonomic levels (*Wright, Tregenza & Hosken, 2008*; *Butchart et al., 2010*; *Magurran et al., 2010*; *Pereira et al., 2010*; *Hoffmann et al., 2011*). Such scarcities confidently offered a viable incentive to regulate the sustainable species variation (*Primack, 2002*; *Duvernell et al., 2008*; *Appeltans et al., 2012*; *Bourlat et al., 2013*; *Leray & Knowlton, 2015*) through advancing genomic protocols to challenge the genetic intimidations such as distraction of local traits, genetic drift and inbreeding effects (*Tallmon, Luikart & Waples, 2004*).

The same conservation obstacle is hypothetically threatening *P. canius* populations in Malaysia, since there is not any comprehensive documentation nor a single initial research on their genetic characterization. Indeed, regarding to their regional significance in Oceania and Southeast Asia (*Usman et al., 2013*), a few regional studies have been merely carried out on basic biological perceptions of *Plotosus canius* including their morphology and fisheries (*Kumar, 2012*; *Usman et al., 2013*), fecundity (*Khan et al., 2002*; *Usman et al., 2013*), feeding behaviour (*Leh, Sasekumar & Chew, 2012*), protein structure (*Prithiviraj & Annadurai, 2012*) and pharmacology (*Prithiviraj, 2014*). Such deficiencies have raised some severe concerns on the level of population structure, genetic variation and the

consequences of genetic differentiation among populations of *P. canius* especially in Malaysia. Thus, this study was performed to genetically characterize *P. canius* through the utilization of the mitochondrial Cytochrome Oxidase I (COI) gene and *Tandanus tandanus* microsatellites, in order to examine the accuracy of employed markers in phylogenetic study, genetic variation assignment, and population genetic structure of *P. canius* in Malaysia.

# MATERIALS AND METHODS

## Sample collection and DNA isolation

Total of 130 catfish samples demonstrating two species of family Plotosidae were directly collected, including of 117 samples of *Plotosus canius* and 13 samples of *Plotosus lineatus*. Sample collection of *P. canius* were performed in five various districts throughout Malaysia including: Negeri Sembilan (NSN), Selangor (SGR), Johor (JHR), Sarawak (SWK) and Sabah (SBH) (Fig. 1) from May–December 2014, while *P. lineatus* samples were only collected from Selangor. Sampling was carried out directly from fishermen in commercial fishing docks of Port Dickson (Negeri Sembilan), Kuala Selangor (Selangor), Kukup (Johor), Bintulu (Sarawak) and Putatan (Sabah). DNA extraction protocol was performed upon sample collection in laboratory via the Wizard® SV Genomic DNA Purification System (Promega, Madison, WI, USA), according to manufacturer's protocol instructions by using roughly 20 mg of specimens.

## PCR amplification and sequencing of mitochondrial DNA

To accurately amplify a 655 bp fragment of mitochondrial DNA, PCR amplification were performed by using C_Fish1 primer set (*Ward et al., 2005*) on the 5′ end of COI gene. The amplification protocol were conducted in an overall volume of 25 $\mu$l at which contained 0.6 $\mu$l of each deoxynucleotide triphosphate (dNTPs), 0.7 $\mu$l Taq polymerase, 1.2 $\mu$l MgCl$_2$, 0.25 $\mu$l of each primers, 2 $\mu$l of concentrated genomic DNA, 2.5 $\mu$l of Taq buffer and 18 $\mu$l of distilled H$_2$O as stated by *Ward et al. (2005)*, with slight modification. The PCR reaction was carried out using an Eppendorf Mastercycler based on the following thermal regime: 2 min of 95 °C initial denaturation step; 35 cycles of 94 °C denaturation step for 30 s, a 54 °C annealing temperature for 45 s and a 72 °C extension period of 1 min; followed by 72 °C final extension step for 10 min and a routine 4 °C final hold (*Ward et al., 2005*). In order to confirm that the PCR reaction generated sufficient amplicon proportions, PCR amplification products were visualized using a 2.0% laboratory grade agarose gel containing 5 $\mu$l GelRed stain. Amplified products were subsequently isolated and purified upon their visualization and documentation. DNA purification from gel was commonly carried out using the Wizard® SV Gel and PCR Clean-up System (Promega, Madison, WI, USA). Purified DNA samples were finally sent to private sector institution (1st Base laboratories Sdn Bhd) for sequencing to generate associated trace files and continuous read lengths intended for genetic and statistical analysis of mitochondrial DNA.

## Statistical analysis of mitochondrial DNA

Trace files were manually end-trimmed using BioEdit software 7.2.5 (*Hall, 1999*) regarding to their homologous section. Afterwards, ClustalX 2.1 (*Thompson et al., 1997*)
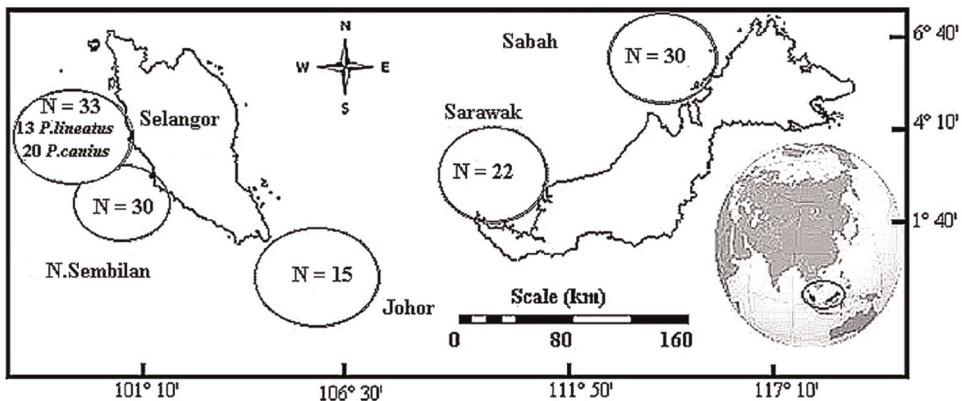

**Figure 1** Sampling sites and sample size (N) diagram of *P. canius* and *P. lineatus* in Malaysia.

was applied to progressively manipulate, align and analyze the DNA sequences. Finally, haplotypes were detected with DnaSP software 5.10.01 (*Librado & Rozas, 2009*) and deposited into BOLDSYSTEM (International Barcode of Life) and GenBank.

To comparatively analyze mitochondrial DNA sequences, MEGA 6 program (*Tamura et al., 2013*) were used intending to understand the phylogenetic outlines of COI gene and species, generate sequence alignment and perform evolutionary analysis. Calculation of the pairwise distance was obtained through 1,000 bootstrap variance estimation and Tamura-Nei model (*Tamura & Nei, 1993*). Moreover, overall mean nucleotide distance of sequences was computed using same configuration at each codon positions separately. Subsequently, construction of phylogenetic tree from the highest grade aligned sequences of *P. canius* and *P. lineatus* was prompted in comparison to one haplotype of African sharp tooth catfish *Clarias gariepinus* (ANGBF8254-12) from Thailand as an outgroup through Neighbor-Joining (NJ) and Maximum Likelihood (ML) methods using a mutual 1,000 replication bootstrap. Next, Minimum Spanning Network (MSN) was computed using PopART (*Bandelt, Forster & Röhl, 1994*) application among obtained sequences of *P. canius*.

Extraction of genetic features from assembly of sequences based on some rudimentary implemented analytical tests was performed through Arlequin software 3.5 (*Excoffier, Laval & Schneider, 2005*). As the most crucial objective was to compute the genetic structure, hierarchical analysis of molecular variance (AMOVA) and pairwise $F_{ST}$ values of chi square test, population differentiation was successively calculated among five populations of *P. canius* in Malaysia. Analysis of molecular variance was carried out using 1,000 permutation to compute distance matrix of sequences, while the same structure were implemented for comparison of all available pair samples and populations through calculation of $F_{ST}$ with 0.05 significance level.

Demographic history was estimated using Tajima's D test (*Tajima, 1989*) and Fu's Fs test (*Fu, 1997*) to test the hypothesis of neutrality of the COI gene. A negative Tajima's D-values might indicate bottleneck, selection or population expansion (*Tajima, 1989*). Mismatch distributions test was also computed to evaluate the hypothesis of recent population expansion or growth (*Rogers & Harpending, 1992*). The distribution is

commonly unimodal in populations that have undergone a recent demographic expansion, and is multimodal in stable or equilibrium populations. All analyses were carried out using ARLEQUIN, while associated mismatch graphs were obtained from DnaSP.

## Microsatellite genotyping

To run measured PCR protocol, optimization of PCR composition were performed at which made positive outcomes based on initial PCR regime using the main protocol for the amplification of *Tandanus tandanus* (*Rourke et al., 2010*) with minor regulation in the amount and concentration of primers, *Taq* DNA polymerase, $MgCl_2$ and dNTPs to enhance the accuracy of the protocol. However, the optimum annealing temperature were practically calculated 55 °C similar as original species. To consciously amplify DNA fragments, PCR amplification were performed by using five cross-amplified microsatellites of *T. tandanus* (*Rourke et al., 2010*; *Rourke & Gilligan, 2010*) on the 5′ end, presented in Table 5. The feasible amplification protocol were conducted in a total volume of 25 μl solution inclosing 0.6 μl of each deoxynucleotide triphosphate (dNTPs), 0.7 μl *Taq* DNA polymerase, 1.2 μl $MgCl_2$, 0.25 μl of each primers, 2 μl of concentrated genomic DNA, 2.5 μl of Taq buffer and 18 μl of distilled $H_2O$ as stated by *Rourke & Gilligan (2010)* with slight modification.

The PCR reaction was carried out using a gradient Eppendorf Mastercycler based on the following thermal adjusted protocol: 2 min of 95 °C initial denaturation step; 35 cycles of 95 °C denaturation step for 30 s, a 55 °C annealing temperature for 45 s and a 72 °C extension period of 1 min; followed by a 72 °C final extension (elongation) step for 10 min and a routine 4 °C final hold (*Rourke et al., 2010*; *Rourke & Gilligan, 2010*). Afterwards, PCR amplification products were visualized using a 4.0% MetaPhor agarose gel containing 5 μl GelRed staining solution. Subsequently, gel images were subjected to microsatellite screening and approximately 15 μl of the florescent label products were packed and sent to First Base Laboratories (private institution) for fragments analysis.

## Genetic analysis of microsatellite markers

In order to verify any null alleles and scoring error, MICROCHECKER 2.2.3 (*Van Oosterhout et al., 2004*) were applied using diploid data obtained from CONVERT software (*Glaubitz, 2004*). GENEPOP 4.2 (*Rousset, 2008*) was employed in order to evaluate the conformity to the "Hardy-Weinberg expectations" (HWE) with 10,000 permutations for test of exact probability. Observed heterozygosity ($H_o$) was estimated using GENALEX 6.5 (*Peakall & Smouse, 2012*). Successively, FSTAT 2.9.3.2 (*Goudet, 1995*) was applied to calculate the expected heterozygosity ($H_e$), with 15,000 permutation and MolKin 3.0 (*Gutiérrez et al., 2005*) to validate the genetic analysis among genetic dataset using Polymorphism Information Content (PIC). Afterwards, ARLEQUIN 3.0 (*Excoffier, Laval & Schneider, 2005*) were used to analyze the genetic configuration, hierarchical AMOVA and pairwise $F_{ST}$ estimations among all five involved populations, using reflection of 95% significance level and 10,000 permutations. Assignment of each individual to their genetic groups ($K$) employing admixture model and its associated

frequency of allelic data was carried out using the STRUCTURE program 2.1 (*Pritchard, Stephens & Donnelly, 2000*). Next, GENECLASS2 2.0 (*Piry et al., 2004*) was implemented to conduct assignment of individuals into the most plausible inheritance group. Finally, probability of current bottleneck was tested using BOTTLENECK software 1.2.02 (*Piry, Luikart & Cornuet, 1999*).

## RESULTS

### Phylogenetic and population analysis inferred from mitochondrial DNA

Among overall number of 130 studied specimens using one COI gene, a full mass of 118 reliable sequences (approximately 91%) were identified. Once the sequencing was completed, the C_Fish1 accordingly confirmed as suitable primer set amplifying in *P. canius* and *P. lineatus* DNA samples. Nevertheless, in some cases there were some uncertain base calls, there was no observation of any stop codons nor any instances of insertion or deletion in sequences. Deficiency of structural stop codon more likely supposed to be associated with every amplified mitochondrial sequences, and all that, alongside with the read length of amplified sequences implies that nuclear sequences initiated from vertebrate mitochondrial DNA are not sequenced. Such occasions are based on the fact that nuclear sequences have typically read lengths less than 600 bp (*Zhang & Hewitt, 1996*). Therefore, the selected COI gene alone was considered for phylogenetic and population structure analysis of *P. canius* and *P. lineatus* in advance.

Preliminary evaluation of verified sequences was mainly generated 20 haplotypes in five populations of *P. canius* and 3 haplotypes in one population of *P. lineatus* (Table 1). Based on the presented data of obtained haplotypes, it can be evidently seen that in *P. canius* samples, KR086940 was found as the most common haplotype in the entire populations from Malaysia, however it was not found in the Negeri Sembilan and the Sabah populations. Moreover, KR086939 was detected as the second common haplotype in *P. canius* populations. The N. Sembilan and the Selangor populations had the most unique haplotypes, while the Johor population had just two shared haplotypes each. In other words, the N. Sembilan and the Selangor population possessed the highest number of identified haplotypes ($n = 6$), while the N. Sembilan and the Sabah populations had five and four haplotypes respectively.

For the 655 available COI nucleotides, 509 sites (roughly 78%) were detected as conserved sites, 146 (22%) as variable sites and 136 (21%) identified as parsim-informative sites. The average nucleotide composition in *P. canius* was 29% T, 27.6% C, 25.2% A and 18.3% G, while the average C + G content of selected positions was calculated as 45.9%, in *P. lineatus* though, calculation was 28, 28.6, 25, 18.4 and 47% respectively. Translation of all 23 haplotypes for conserved 655 bp fragment was produced 165 amino acids, which presented no signal of pseudogene in their structure. However, the sample size noticeably varied, ranging from 13–30 in collected samples of *P. lineatus* and *P. canius* from five different districts in Malaysia; in regard to the fact that original sample sizes were moderate and some sequences were failed, the actual sample collection

**Table 1** Overview of haplotypes, their sampling sites and accession numbers.

| Species | BOLDSYSTEM index | GenBank accession number | Sampling site |
|---|---|---|---|
| *Plotosus lineatus* | NUPM017-14 | KP258659 | Selangor |
| | NUPM016-14 | KP258657 | |
| | NUPM015-14 | KP258658 | |
| *Plotosus canius* | NUPM001-14 | KP258648 | Negeri Sembilan |
| | NUPM002-14 | KP258651 | |
| | NUPM006-14 | KP258655 | |
| | NUPM023-15 | KR086935 | |
| | NUPM003-14 | KP258650 | Sabah |
| | NUPM004-14 | KP258649 | |
| | NUPM005-14 | KP258656 | |
| | NUPM022-15 | KR086936 | |
| | NUPM007-14 | KP258654 | Selangor |
| | NUPM008-14 | KP258653 | |
| | NUPM009-14 | KP258652 | |
| | NUPM020-15 | KR086938 | |
| | NUPM021-15 | KR086937 | |
| | NUPM010-14 | KP221601 | Sarawak |
| | NUPM011-14 | KP221602 | |
| | NUPM012-14 | KP221603 | |
| | NUPM013-14 | KP221604 | |
| | NUPM014-14 | KP221605 | |
| | | | Selangor |
| | NUPM018-15 | KR086940 | Sarawak |
| | | | Johor |
| | NUPM019-15 | KR086939 | Johor |
| | | | Negeri Sembilan |

is in the desired range recommended by *Zhang et al. (2010)*. Basically, in order to detect approximately 80% genetic variation, a collection of 31.9–617.8 samples could be needed, although it is far from real laboratory and field work measures; hence, it is suggested that at least 10 sample might be desirably sufficient to accurately identify the genetic variability in real phylogenetic studies (*Zhang et al., 2010*). An overview of most crucial outcomes in polymorphism analysis (Table 2) indicated that the number of variable sites are moderately fluctuating from 2 in *P. lineatus* to 54 in *P. canius* samples from Sarawak, While the degree of nucleotide diversity was relatively low (0.00067–0.0391) and as its consequence, the level of polymorphism and genetic variation in populations reasonably presented small portion. Besides, the degree of haplotypes diversity waved from 0.395 (Sabah) to 0.771 (Sarawak).

The Tamura-Nei pairwise distance matrix (Table 3) indicated a comparatively high overall interspecies pairwise divergence of 25.2%, while the least interspecific distance

Table 2 Summary of 23 observed mitochondrial DNA haplotypes and their distribution, nucleotide diversity, number of haplotypes, haplotype diversity and number of polymorphic sites.

| Haplotype | P. canius | | | | | P. lineatus |
| --- | --- | --- | --- | --- | --- | --- |
| GenBank accession number | Selangor $n = 20$ | Negeri Sembilan $n = 18$ | Johor $n = 15$ | Sabah $n = 30$ | Sarawak $n = 22$ | Selangor $n = 13$ |
| KP258659 | | | | | | 7.8* |
| KP258657 | | | | | | 15.3* |
| KP258658 | | | | | | 76.9* |
| KP258648 | | 5.6* | | | | |
| KP258651 | | 16.6* | | | | |
| KP258655 | | 11.1* | | | | |
| KR086935 | | 5.6* | | | | |
| KP258650 | | | | 3.3* | | |
| KP258649 | | | | 3.3* | | |
| KP258656 | | | | 76.6* | | |
| KR086936 | | | | 16.8* | | |
| KP258654 | 5* | | | | | |
| KP258653 | 10* | | | | | |
| KP258652 | 5* | | | | | |
| KR086938 | 5* | | | | | |
| KR086937 | 10* | | | | | |
| KP221601 | | | | | 4.5* | |
| KP221602 | | | | | 31.8* | |
| KP221603 | | | | | 13.6* | |
| KP221604 | | | | | 4.5* | |
| KP221605 | | | | | 9* | |
| KR086940 | 65* | | 60* | | 36.6* | |
| KR086939 | | 61.1* | 40* | | | |
| Nucleotide diversity (Pi JC) | 0.00457 | 0.00184 | 0.00306 | 0.00134 | 0.0391 | 0.00067 |
| Number of haplotypes | 6 | 5 | 2 | 4 | 6 | 3 |
| Haplotype diversity (Hd) | 0.642 | 0.614 | 0.667 | 0.395 | 0.771 | 0.410 |
| Number of polymorphic site | 14 | 5 | 3 | 4 | 57 | 2 |

Note:
* Haplotype frequencies in each population are presented as percentage.

was 0.2%. The Tamura-Nei intraspecific distance however, ranged from 0.2–9.7% between *P. canius* from Sarawak and Selangor. Nevertheless, the majority of *P. canius* pairwise distances displayed low levels of conspecific divergence roughly around 1%. The greatest genetic differences was observed between the Selangor and Sarawak (KR086937–KP221604) samples (9.7%), which is moderately reasonable due to their geographical distance. The next significantly high variances was detected between the Sabah-Sarawak

**Table 3** Pairwise Tamura-Nei genetic distance in 23 employed haplotypes of *P. canius* and *P. lineatus*.

| | 1 | 2 | 3 | 4 | 5 | 6 | 7 | 8 | 9 | 10 | 11 | 12 | 13 | 14 | 15 | 16 | 17 | 18 | 19 | 20 | 21 | 22 | 23 |
|---|---|---|---|---|---|---|---|---|---|---|---|---|---|---|---|---|---|---|---|---|---|---|---|
| 1 KP221604 | | | | | | | | | | | | | | | | | | | | | | | |
| 2 KP221603 | 0.002 | | | | | | | | | | | | | | | | | | | | | | |
| 3 KP221605 | 0.003 | 0.002 | | | | | | | | | | | | | | | | | | | | | |
| 4 KP258655 | 0.091 | 0.089 | 0.087 | | | | | | | | | | | | | | | | | | | | |
| 5 KP258654 | 0.093 | 0.091 | 0.089 | 0.002 | | | | | | | | | | | | | | | | | | | |
| 6 KR086940 | 0.091 | 0.089 | 0.087 | 0.089 | 0.002 | | | | | | | | | | | | | | | | | | |
| 7 KP258653 | 0.090 | 0.088 | 0.086 | 0.002 | 0.003 | 0.002 | | | | | | | | | | | | | | | | | |
| 8 KR086937 | 0.097 | 0.094 | 0.092 | 0.005 | 0.006 | 0.005 | 0.006 | | | | | | | | | | | | | | | | |
| 9 KP258652 | 0.096 | 0.094 | 0.092 | 0.006 | 0.008 | 0.006 | 0.008 | 0.005 | | | | | | | | | | | | | | | |
| 10 KR086938 | 0.089 | 0.087 | 0.084 | 0.002 | 0.003 | 0.002 | 0.003 | 0.006 | 0.006 | | | | | | | | | | | | | | |
| 11 KP221601 | 0.091 | 0.089 | 0.087 | 0.003 | 0.005 | 0.003 | 0.005 | 0.008 | 0.009 | 0.005 | | | | | | | | | | | | | |
| 12 KR086936 | 0.089 | 0.087 | 0.084 | 0.005 | 0.006 | 0.005 | 0.006 | 0.009 | 0.011 | 0.006 | 0.008 | | | | | | | | | | | | |
| 13 KP258650 | 0.090 | 0.088 | 0.086 | 0.006 | 0.008 | 0.006 | 0.008 | 0.011 | 0.012 | 0.008 | 0.009 | 0.002 | | | | | | | | | | | |
| 14 KP258649 | 0.091 | 0.089 | 0.087 | 0.006 | 0.008 | 0.006 | 0.008 | 0.011 | 0.012 | 0.008 | 0.009 | 0.002 | 0.003 | | | | | | | | | | |
| 15 KP258656 | 0.093 | 0.091 | 0.089 | 0.005 | 0.006 | 0.005 | 0.006 | 0.009 | 0.011 | 0.006 | 0.008 | 0.003 | 0.005 | 0.005 | | | | | | | | | |
| 16 KP258648 | 0.087 | 0.084 | 0.082 | 0.006 | 0.008 | 0.006 | 0.008 | 0.011 | 0.012 | 0.008 | 0.009 | 0.005 | 0.006 | 0.006 | 0.005 | | | | | | | | |
| 17 KR086935 | 0.087 | 0.084 | 0.082 | 0.006 | 0.008 | 0.006 | 0.008 | 0.011 | 0.012 | 0.008 | 0.009 | 0.005 | 0.006 | 0.006 | 0.005 | 0.000 | | | | | | | |
| 18 KP258651 | 0.089 | 0.087 | 0.084 | 0.005 | 0.006 | 0.005 | 0.006 | 0.009 | 0.011 | 0.006 | 0.008 | 0.003 | 0.005 | 0.005 | 0.003 | 0.002 | 0.002 | | | | | | |
| 19 KR086939 | 0.089 | 0.087 | 0.084 | 0.005 | 0.006 | 0.005 | 0.006 | 0.009 | 0.011 | 0.006 | 0.008 | 0.003 | 0.005 | 0.005 | 0.003 | 0.002 | 0.002 | 0.000 | | | | | |
| 20 KP221602 | 0.078 | 0.076 | 0.078 | 0.028 | 0.029 | 0.028 | 0.029 | 0.032 | 0.032 | 0.026 | 0.028 | 0.026 | 0.027 | 0.027 | 0.026 | 0.024 | 0.024 | 0.022 | 0.022 | | | | |
| 21 KP258657 | 0.241 | 0.239 | 0.235 | 0.238 | 0.240 | 0.238 | 0.240 | 0.246 | 0.246 | 0.235 | 0.245 | 0.240 | 0.242 | 0.237 | 0.240 | 0.237 | 0.237 | 0.241 | 0.241 | 0.252 | | | |
| 22 KP258658 | 0.237 | 0.235 | 0.231 | 0.235 | 0.237 | 0.235 | 0.237 | 0.243 | 0.243 | 0.232 | 0.241 | 0.236 | 0.239 | 0.234 | 0.236 | 0.234 | 0.234 | 0.237 | 0.237 | 0.248 | 0.002 | | |
| 23 KP258659 | 0.237 | 0.235 | 0.231 | 0.235 | 0.237 | 0.235 | 0.237 | 0.243 | 0.243 | 0.232 | 0.241 | 0.236 | 0.239 | 0.234 | 0.236 | 0.234 | 0.234 | 0.237 | 0.237 | 0.248 | 0.003 | 0.002 | |

pair (KP258656–KP221604) and Negeri Sembilan-Sarawak (KP258654–KP221604), although in later occasion, distance between Sarawak-Sabah sites is extraordinarily closer than Sarawak-Negeri Sembilan.

Phylogenetic analysis using the 23 haplotypes of the genus *Plotosus* and one haplotype of *C. gariepinus* showed monophyletic status between *P. canius* and *P. lineatus* via the Neighbour-Joining method (Fig. 2). As stated, the three haplotypes from Sarawak population formed a basal clade for *P. canius* using Neighbour-Joining algorithm. Moreover, constructed topology precisely proved the pairwise genetic distances of shared haplotypes, highlighting that KR086940 (SGR, SWK, JHR) have the lowest distance to KP258654 (SGR) and highest to KP221604 (SWK), while KR086939 (JHR, NSN) have the greatest divergence from KP221604 (SWK) and smallest amount from KP258648 (NSN), as their subdivision clades illustrated.

Similarly, application of Maximum Likelihood algorithm (Fig. 3) have been subsequently confirmed the calculation of pairwise genetic distances among sequences of *P. canius* and *P. lineatus* with exception of some negligible changes in topology of clades and branches.

The MSN of 20 haplotypes of *P. canius* (Fig. 4) in Malaysia presented more haplotype variability in Sarawak and Negeri Sembilan populations with six and five haplotypes respectively. Indeed, the Sarawak and Negeri Sembilan sequences illustrated a fairly high diversity, while the two haplotypes of Johor possessed the lowest variability. However, the phylogram revealed two relatively irrefutable geographical clades (Sarawak and Negeri Sembilan), occurrence of mix haplotypes with other clades indicating that no accurate geographical genetic structure have been certainly detected in studied populations of *P. canius*. Analysis of potential geographical clades have been similarly suggested that all populations are moderately mixed except in Sarawak and Negeri Sembilan. Although there are some possible clades, it was not precisely feasible to clustering the population based on their geographical divisions due to existence of exclusively one connecting mutational steps for most sequences. Hence, analysis was not capable of showing precise separation of geographical clades.

Analysis of population differentiation inferred from pairwise distance of $F_{ST}$ and Chi-square among studied populations of *P. canius* is displayed in Table 4. Significant genetic variations were detected in all assessments within *P. canius* sequences and two considered species of eel-tail catfishes ($P < 0.001$). However, there were significant distance diversity in genetic variations of almost entire evaluations among populations of Malaysia especially between the Sabah and the Negeri Sembilan populations. As expected, the most diversity were identified between *P. canius* populations from the Negeri Sembilan and the Sabah with the rate of 0.62504, which basically means they are nearly genetically divided due to their distance and subsequent decrease in gene flow. However, the lowest genetic distance was detected between Selangor cluster and the Johor clade by the $F_{ST}$ values of 0.05417. Hence, it was considered that maximum sharing of genetic material occurred between Johor and Selangor Populations, while the minimum genetic similarity identified among Sabah and Negeri Sembilan sequences.

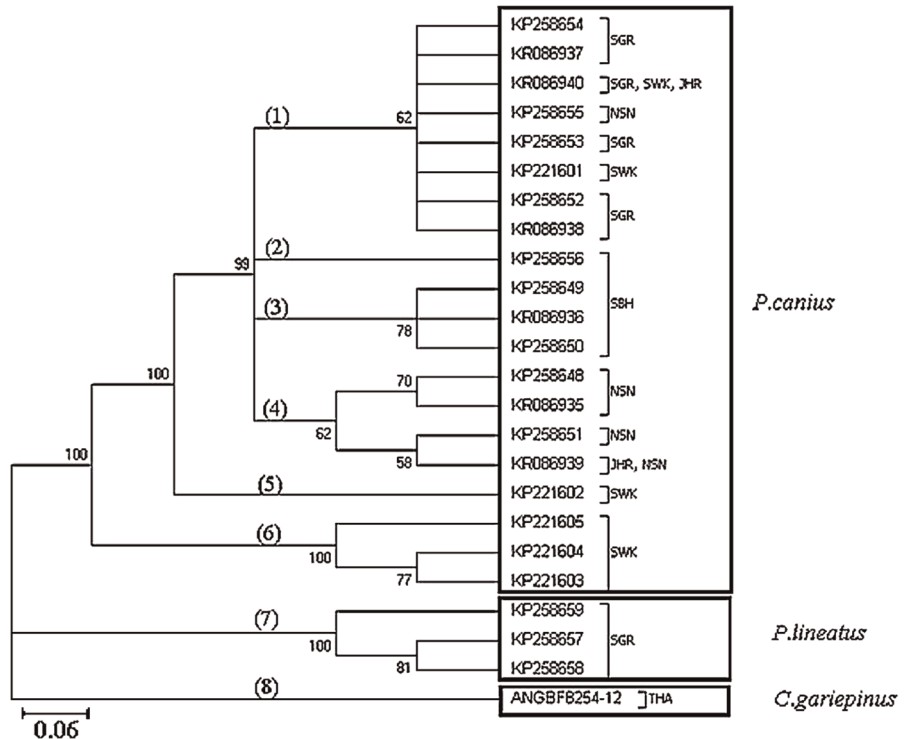

**Figure 2** Summary of Neighbour-Joining relationship in 24 employed sequences of *P. canius*, *P. lineatus* and *C. gariepinus* (clades have been indicated by bold numbers in round brackets).

Hierarchical statistics of AMOVA (Table 5) was clearly suggested that roughly 36% of experimental deviations were inter-population variations, while within population variations were merely responsible for approximately 64% of overall differentiation.

The illustration of mismatch distribution (Fig. 5) have shown multimodal pattern for *P. canius* populations in Malaysia. Moreover, Tajima's *D* test was negative for Selangor, Sabah and N. Sembilan populations, but not significant (Table 6). Besides, Fu's test showed negative signal for Selangor population, which wasn't similarly significant. These statistics suggest that populations of *P. canius* didn't possibly experience demographic expansion for a long period of time in Malaysia. However, comparison of $\theta_1$ and $\theta_0$ parameters for all populations indicates a relatively slow growth rate in female populations except in Sabah.

## Population genetic results inferred from genotyping analysis

Fragment analysis were estimated DNA band sizes in *P. canius* that are illustrated alongside with size range in original species (*Tandanus tandanus*), the sequence of each primers and associate annealing temperatures in Table 7.

Genotyping results did not found any signal of null allele nor large allele failure, hence nor scoring inaccuracies due to stuttering. All the five microsatellites loci showed accurate and successful amplification in all populations, although heterozygous alleles were not found in all populations. One microsatellite loci (Tan 3-27), was evidently monomorphic in all populations while Tan 1-2, Tan 1-10, Tan 1-7 and Tan 3-28 were polymorphic in
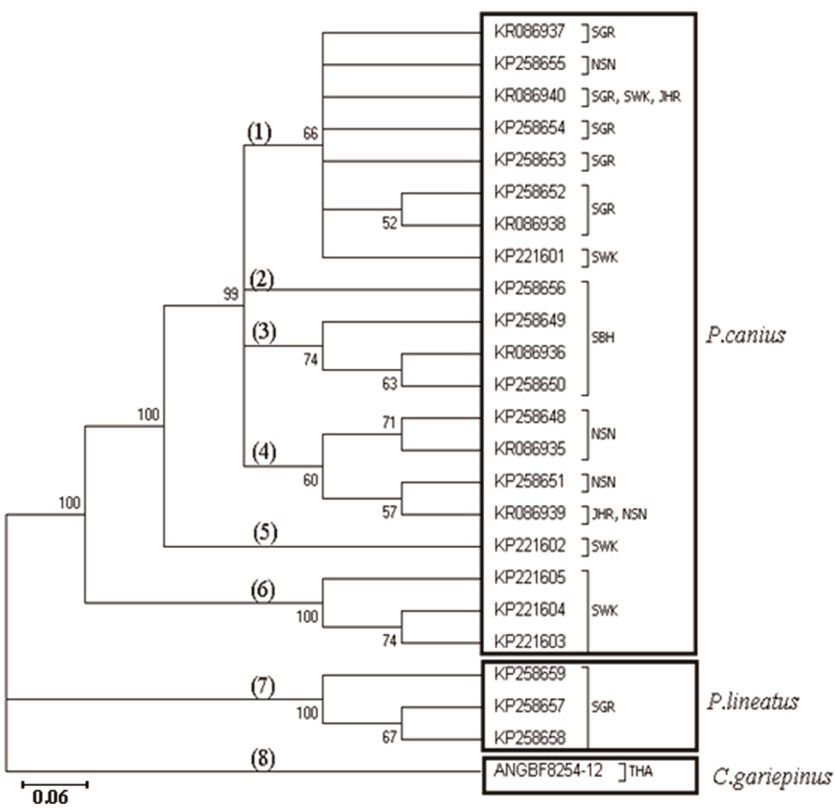

**Figure 3** Summary of maximum likelihood relationship in 24 employed sequences of *P. canius*, *P. lineatus* and *C. gariepinus* (clades have been indicated by bold numbers in round brackets).

at least one population. After implementing the sequential Bonferroni adjustment (*Rice, 1989*), only four out of the 50 (8%) loci pairs were significant for linkage disequilibrium ($P < 0.05$). Thus, all the five microsatellites loci were considered useful for genetic applications based on the absence of consistent linkage disequilibrium in locus pairs among the studied populations.

Furthermore, after Bonferroni adjustment, nine out of the 25 (36%) microsatellite loci still showed deviation from Hardy-Weinberg Equilibrium (HWE), which might be owing to heterozygote deficiency effects (Table 8). Heterozygote deficiency could be caused by population structuring, null alleles or inbreeding (*Brook et al., 2002*).

Data from Table 8 showed that six out of nine deviations were related to Tan 1-7 and Tan 3-28 among all the five *P. canius* populations. The fact that the two loci did not show any signal of deviation from HWE in three populations may imply the probability that the estimated deviations could have been originated from either the occurrence of uncertain structure or inbreeding among these three population divisions (*Pritchard, Stephens & Donnelly, 2000*). Fairly high level of consistency was detected toward inclusion or exclusion of Tan 1-7 and Tan 3-28, accordingly these loci have been retained for further analysis. *Fis* ($P < 0.05$) estimations have been considerably diverse from zero, except in locus Tan 1-2 from Sabah. This alongside with substantial departure from HWE indicates the damaging effect of heterozygote deficiency within associated populations.

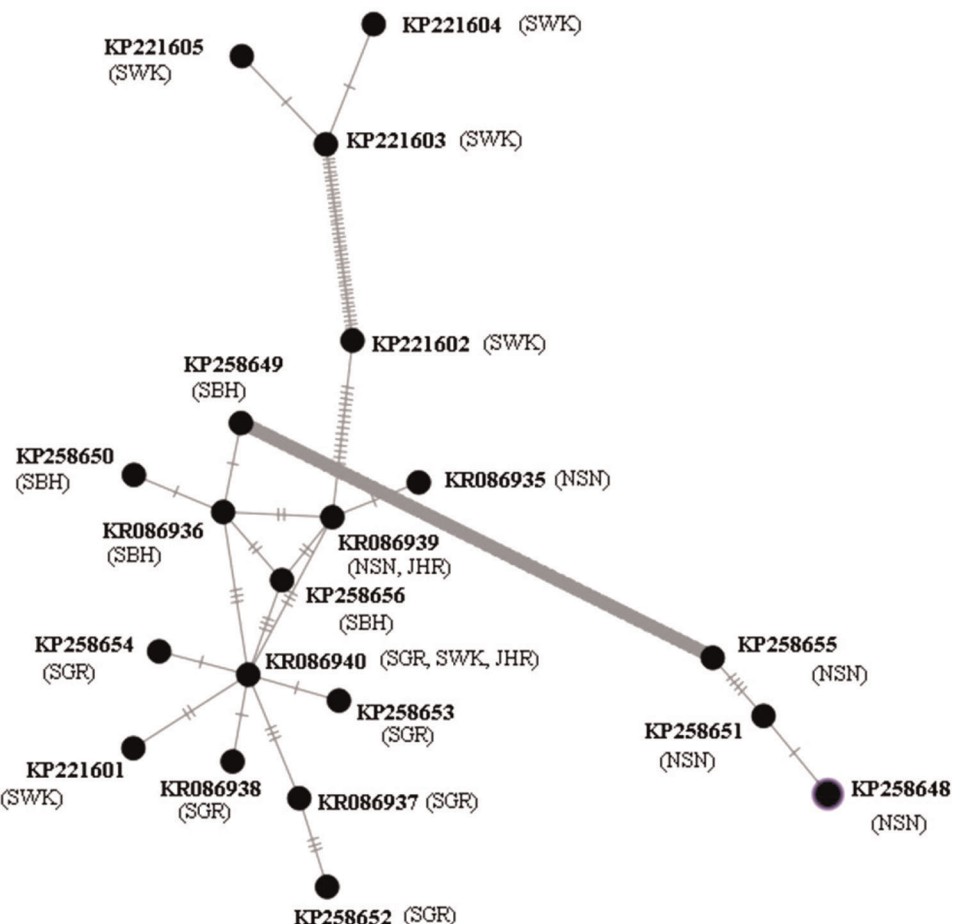

**Figure 4** MSN of 20 haplotypes of *P. canius*.

**Table 4** Population pairwise ($F_{ST}$) values of chi square test for population variation originated with 1,000 permutations.

|  | *P. canius* SBH | *P. canius* JHR | *P. canius* SGR | *P. canius* NSN | *P. canius* SWK |
|---|---|---|---|---|---|
| *P. canius* SBH | 0.00000 |  |  |  |  |
| *P. canius* JHR | 0.60156 | 0.00000 |  |  |  |
| *P. canius* SGR | 0.43390 | 0.05417 | 0.00000 |  |  |
| *P. canius* NSN | 0.62504 | 0.44097 | 0.09806 | 0.00000 |  |
| *P. canius* SWK | 0.41533 | 0.27777 | 0.29359 | 0.31333 | 0.00000 |

**Table 5** Hierarchical AMOVA in *P. canius*.

| Source of variation | Degree of freedom | Sum of squares | Variance components | Percentage of variation |
|---|---|---|---|---|
| Among populations | 4 | 77.873 | 3.09472 | 35.55 |
| Between populations | 18 | 100.969 | 5.60941 | 64.45 |
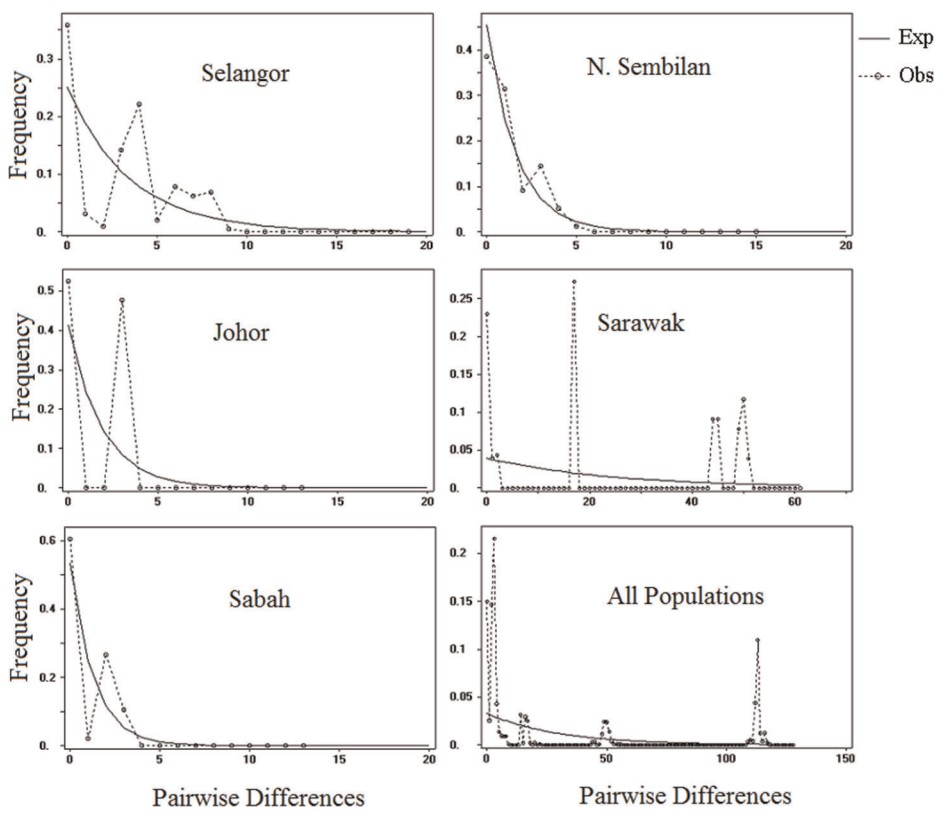

**Figure 5** Mismatch distribution of *P. canius* for different geographical regions and whole population.

**Table 6** Tajima's *D* and Fu's *Fs*, corresponding *P*-value, and mismatch distribution parameter estimates (significance level: $P < 0.01$).

| Species | Tajima's *D* | | Fu's *Fs* | | Mismatch distribution | | |
|---|---|---|---|---|---|---|---|
| | *D* | *P* | *Fs* | *P* | $\tau$ (95% CI) | $\theta_0$ | $\theta_1$ |
| Selangor | −0.89 | 0.20 | −0.01 | 0.53 | 10.98 | 1.64 | 3.44 |
| N. Sembilan | −0.84 | 0.23 | 0.72 | 0.61 | 4.18 | 0.334 | 0.62 |
| Johor | 1.63 | 0.96 | 3.63 | 0.94 | 26.17 | 0.99 | 1.51 |
| Sabah | −0.35 | 0.42 | 0.08 | 0.49 | 0.74 | 0.001 | 99,783 |
| Sarawak | 2.24 | 0.99 | 15.14 | 1.00 | 58.39 | 26.44 | 53.03 |
| All populations | 1.42 | 0.38 | 6.08 | 0.33 | 22.39 | 8.74 | 51,632 |

However, the positive calculated estimations could be translated as decrease in heterozygous levels among offspring in population, mostly owning to non-random mating and its subsequent inbreeding. On the other hand, negative *Fis* estimates might be indication of increasing in heterozygosity level, which could usually occur as a result of random mating system, hence genes should be probably more different (*Pritchard, Stephens & Donnelly, 2000*).

Analysis of population genetic inferred from molecular coancestry information (Table 9) revealed that PIC of the applied microsatellites varied from 19.86–73.99.

**Table 7** Five engaged primer sets and their associated size and temperature in *T. tandanus* and *P. canius*.

| Primer | Sequence | Size *T. tandanus* | Size *P. canius* | Annealing temperature |
|--------|----------|--------------------|-----------------|------------------------|
| Tan 1-2 | F: 5′CCGACTGTCAGTGAAAAGGAG3′<br>R: 5′AGGGTCTGGGAGTGAATGAG3′ | 216–244 | 349–385 | 55 °C |
| Tan 1-7 | F: 5′TGTATGGAGCTACTAACAAAACAGG3′<br>R: 5′TACTCCAGCCCTGAAGGTG3′ | 181–227 | 114–125 | 55 °C |
| Tan 1-10 | F: 5′CCTGATTTCTCTCCCAAGG3′<br>R: 5′AGAAAGGTGGTGCATGTGTG3′ | 298–310 | 91–97 | 55 °C |
| Tan 3-27 | F: 5′TGTGGAAGGTTGGGGTTATG3′<br>R: 5′CGTGATCATGCAAACAGATG3′ | 215–269 | 167–168 | 55 °C |
| Tan 3-28 | F: 5′CCCCATTTGCTTTTTCTCTG3′<br>R: 5′TGTTGAAAGCGGCATGTTAG3′ | 289–301 | 280–299 | 55 °C |

However, loci with numerous allele numbers and a PIC value of 1 are considered as highly polymorphic and thus most desirable, lowest rates are also slightly informative (if and only if PIC > 0) (*Botstein et al., 1980*). Besides, the arithmetic values of heterozygosity (0.2235–0.8357) confirmed the consistent application of all the five microsatellite loci in population genetic study of *P. canius* in Malaysia.

Hierarchical results of microsatellites showed that approximately 64% of experimental variations were originated from inter–population variations, while within individuals variations were only accountable for roughly 28.5% of overall differentiation (Table 10).

$F_{ST}$ plot estimations of all involved microsatellites noticeably illustrated that all pairwise calculations presented a fairly high differentiations among populations ranging from 0.29711 between populations of Selangor and Negeri Sembilan to 0.80500 between populations of Sarawak and Johor (Table 11). The current microsatellite experiment showed that the highest genetic differentiation was between the Johor population and the other populations from Peninsular Malaysia and Borneo. Indeed, the Johor population showed strong deviation from other collected populations, while displayed relatively low intra-population genetic variation (Table 11), suggesting that the Johor population was less connected to the others during a sizeable period of its evolutionary phase. Likewise, the Sarawak population in the Southwest region of the South China Sea had high $F_{ST}$ values between other populations, signifying the restricted gene flow between the Sarawak population and the other populations. The differentiation level among populations of Negeri Sembilan and Selangor ($F_{ST} = 0.29711$, $P < 0.05$) showed relatively lower values, even in comparison with their neighbouring populations, theoretically demonstrating the populations that endured inbreeding or genetic drift since their isolation from other populations. Similarly, the Negeri Sembilan and Sabah ($F_{ST} = 0.39244$, $P < 0.05$) populations also showed small but significant variances in relation to their close neighbour populations.

Levels of genetic variations seem to be widely fluctuated between *P. canius* populations owning to $H_e$ and *Ar* oscillation on which $H_e$ extending from 0.0000–0.6769 (Selangor) and *Ar* varied from 1–3 (Negeri Sembilan and Selangor). Obviously, two significant

**Table 8** Genetic variation at 5 microsatellite as of five populations of *P. canius* in Malaysia.

| Locus | N. Sembilan | Sabah | Selangor | Sarawak | Johor | Total* Mean |
|---|---|---|---|---|---|---|
| $N$ | 30 | 30 | 20 | 22 | 15 | 117 |
| **Tan 1-2** | | | | | | |
| $N_a$ | 1 | 2 | 1 | 1 | 2 | 4 |
| Ar | 1.000 | 2.000 | 1.000 | 1.000 | 2.000 | 3.714 |
| $H_o$ | 0.0000 | 0.0000 | 0.0000 | 0.0000 | 0.6000 | 0.0769 |
| $H_e$ | 0.0000 | 0.4994 | 0.0000 | 0.0000 | 0.4345 | 0.5974 |
| $Fis$ | – | 0.000 | – | – | −0.400 | |
| HW | – | 1.0000 | – | – | 0.1791 | |
| **Tan 1-7** | | | | | | |
| $N_a$ | 2 | 2 | 3 | 2 | 1 | 6 |
| Ar | 2.000 | 2.000 | 3.000 | 2.000 | 1.000 | 5.844 |
| $H_o$ | 1.000 | 0.0000 | 0.6000 | 0.0000 | 0.0000 | 0.3590 |
| $H_e$ | 0.5085 | 0.4994 | 0.6769 | 0.4947 | 0.0000 | 0.7727 |
| $Fis$ | −1.000 | 1.000 | 0.116 | 1.000 | – | |
| HW | 0.0000 | 1.0000 | 0.5856 | 1.0000 | – | |
| **Tan 1-10** | | | | | | |
| $N_a$ | 1 | 1 | 1 | 1 | 2 | 3 |
| Ar | 1.000 | 1.000 | 1.000 | 1.000 | 2.000 | 2.564 |
| $H_o$ | 0.0000 | 0.0000 | 0.0000 | 0.0000 | 0.4000 | 0.0513 |
| $H_e$ | 0.0000 | 0.0000 | 0.0000 | 0.0000 | 0.3310 | 0.3472 |
| $Fis$ | – | – | – | – | −0.217 | |
| HW | – | – | – | – | 0.5395 | |
| **Tan 3-27** | | | | | | |
| $N_a$ | 1 | 1 | 1 | 1 | 1 | 2 |
| Ar | 1.000 | 1.000 | 1.000 | 1.000 | 1.000 | 1.988 |
| $H_o$ | 0.0000 | 0.0000 | 0.0000 | 0.0000 | 0.0000 | 0.0000 |
| $H_e$ | 0.0000 | 0.0000 | 0.0000 | 0.0000 | 0.0000 | 0.2242 |
| $Fis$ | – | – | – | – | – | |
| HW | – | – | – | – | – | |
| **Tan 3-28** | | | | | | |
| $N_a$ | 3 | 2 | 3 | 2 | 1 | 8 |
| Ar | 3.000 | 2.000 | 3.000 | 2.000 | 1.000 | 7.592 |
| $H_o$ | 0.4667 | 1.0000 | 0.3000 | 0.0000 | 0.0000 | 0.4274 |
| $H_e$ | 0.6169 | 0.5885 | 0.4769 | 0.4947 | 0.0000 | 0.8393 |
| $Fis$ | 0.247 | −1.000 | 0.377 | 1.000 | – | |
| HW | 0.5862 | 0.0000 | 0.5771 | 1.0000 | – | |

**Note:**
Sample size ($N$), Number of alleles ($N_a$), allele richness (Ar), ($H_o$), ($H_e$), inbreeding coefficient ($Fis$) ($P < 0.05$ symbolic accustomed nominal level (5%) 0.000042, and Hardy-Weinberg expectation (disequilibrium) (HW).

clusters could be seen among established populations of *P. canius* in this study, one cluster with low allelic richness and $H_e$ estimations (Johor samples using locus Tan 1-2 and locus Tan 1-10), and another cluster with relatively acceptable $H_e$ and allelic richness (the other four populations using locus Tan 3-28 and locus Tan 1-7). However, the genetic variation

**Table 9** Analysis of population genetic using molecular coancestry information.

| Microsatellite | $N_a$ | Heterozigosity* | PIC (%) | Effective allele no. |
|---|---|---|---|---|
| Tan 1-2 | 4 | 0.5949 | 54.12 | 2.47 |
| Tan 1-7 | 6 | 0.7694 | 73.99 | 4.34 |
| Tan 1-10 | 3 | 0.3457 | 30.11 | 1.53 |
| Tan 3-27 | 2 | 0.2235 | 19.86 | 1.29 |
| Tan 3-28 | 8 | 0.8357 | 81.71 | 6.09 |

Notes:
Number of alleles ($N_a$) and PIC.
* Heterozygosity was estimated as arithmetic mean of expected and $H_o$.

**Table 10** Hierarchical AMOVA in *P. canius*.

| Source of variation | Sum of squares | Variance components | Variation % |
|---|---|---|---|
| Among populations | 191.456 | 1.02366 | 63.77361 |
| Among individuals within populations | 79.039 | 0.12422 | 7.73898 |
| Within individuals | 53.500 | 0.45726 | 28.48741 |

in the current study was highly reliant on microsatellites and their sequences as the engaged loci did not specifically develop for *P. canius*. Moreover, allelic frequencies among virtually each combination of population pairs showed highly significant differentiation ($F_{ST} < 0.05$) (Table 11), implying that gene flow might be highly restricted among studied populations.

Sampled populations of *P. canius* were basically distributed into five minor clusters using Bayesian analysis. Consequently, the initial highest membership value (q) of the studied populations including Negeri Sembilan, Sabah, Selangor, Sarawak, Johor was estimated as 0.941, 0.983, 0.968, 0.988, and 0.986 respectively (Table 12). The application of STRUCTURE program subsequently illustrated 4 major *K* (isolated clusters) (Fig. 6). Regarding the fact that assessing the expected value of *K* might not be straightforward (*Evanno, Regnaut & Goudet, 2005*), Bayesian structure analysis of the current study revealed the highest probability of *K* for *P. canius* in *K* = 2. The four estimated clusters were included Cluster 1: Johor, Cluster 2: N. Sembilan and Selangor, Cluster 3: Sabah, and Cluster 4: Sarawak (Fig. 7).

Population assignment outcomes evidently revealed that almost all individuals were assigned to their original populations with the probability rate of $P > 0.05$ (Table 13). However, the estimation of individual assignment to their populations with the same probabilities revealed the closer rates in comparison levels to other sampling sites. For instance, the Negeri Sembilan population had a relatively closer assignment ratio to the Selangor population.

Analysis of population bottleneck did not identified substantiating signals of recent population drop in all populations studied using the two phase model (T.P.M) estimations (Table 14). Furthermore, calculation of the infinite-allele model (I.A.M) comprehensively implied that there was no bottleneck evidence among the studied

**Table 11 Pairwise $F_{ST}$ estimations through _P. canius_ populations generated from five microsatellites loci and inclusion of five populations.** All calculations were fairly significant ($P < 0.05$) using 10,000 permutations.

|   |             | N. Sembilan | Sabah   | Selangor | Sarawak | Johor   |
|---|-------------|-------------|---------|----------|---------|---------|
| 1 | N. Sembilan | 0.00000     |         |          |         |         |
| 2 | Sabah       | 0.39244     | 0.00000 |          |         |         |
| 3 | Selangor    | 0.29711     | 0.41039 | 0.00000  |         |         |
| 4 | Sarawak     | 0.71934     | 0.68086 | 0.72437  | 0.00000 |         |
| 5 | Johor       | 0.73211     | 0.68561 | 0.74834  | 0.80500 | 0.00000 |

**Table 12 Membership ratio estimated for each population of _P. canius_.**

| Populations | Cluster membership | | | | |
|-------------|-------|-------|-------|-------|-------|
|             | 1     | 2     | 3     | 4     | 5     |
| N. Sembilan | 0.005 | 0.041 | 0.941 | 0.010 | 0.004 |
| Sabah       | 0.005 | 0.004 | 0.005 | 0.983 | 0.003 |
| Selangor    | 0.005 | 0.968 | 0.019 | 0.005 | 0.004 |
| Sarawak     | 0.003 | 0.003 | 0.003 | 0.003 | 0.988 |
| Johor       | 0.985 | 0.004 | 0.004 | 0.004 | 0.003 |

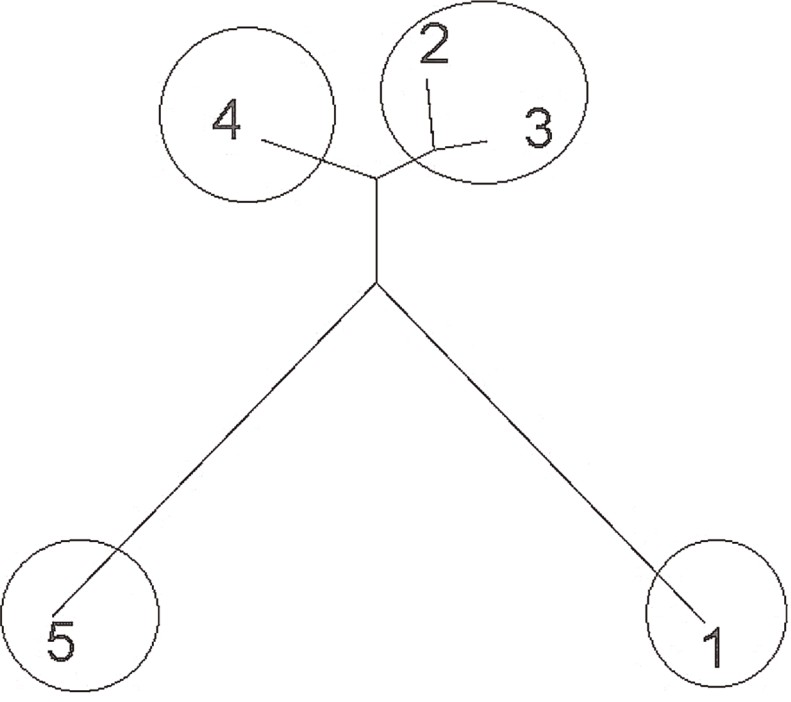

**Figure 6** Tree plot scheme of five engaged populations of _P. canius_.

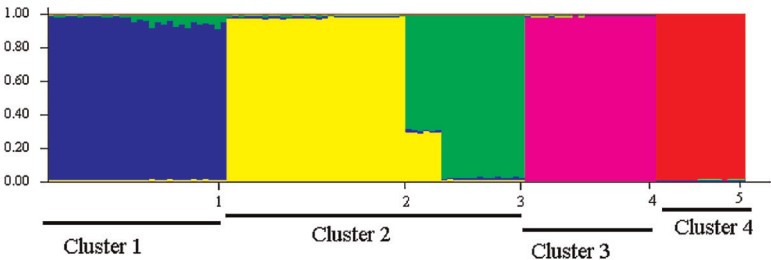

**Figure 7** Population structure of five *P. canius* populations in Malaysia.

**Table 13** Population assignment based upon five microsatellite loci frequencies in *P. canius*.

| Assigned population | CA% | Original location | | | | |
| --- | --- | --- | --- | --- | --- | --- |
| | | N. Sembilan (*n* = 30) | Sabah (*n* = 30) | Selangor (*n* = 20) | Sarawak (*n* = 22) | Johor (*n* = 15) |
| N. Sembilan | 100 | 4.219 | 70.330 | 35.873 | 117.792 | 92.839 |
| Sabah | 100 | 70.143 | 4.338 | 58.286 | 132.093 | 96.994 |
| Selangor | 100 | 35.937 | 58.537 | 4.715 | 101.049 | 83.156 |
| Sarawak | 100 | 115.433 | 129.921 | 98.626 | 3.133 | 101.070 |
| Johor | 100 | 88.554 | 92.895 | 78.807 | 99.143 | 2.761 |

Note:
CA, correct assignment.

**Table 14** *P* values originated from bottleneck analysis within five populations of *P. canius*.

| | I.A.M | T.P.M | | | S.M.M | Mode Shift |
| --- | --- | --- | --- | --- | --- | --- |
| | | 60 | 70 | 80 | | |
| N. Sembilan | 0.22672 | 0.24435 | 0.26030 | 0.26596 | 0.28595 | Y |
| Sabah | 0.07656 | 0.09247 | 0.09295 | 0.09683 | 0.10790 | Y |
| Selangor | 0.28119 | 0.31416 | 0.34221 | 0.32656 | 0.64363 | Y |
| Sarawak | 0.17976 | 0.21249 | 0.21558 | 0.20907 | 0.24057 | Y |
| Johor | 0.18105 | 0.21600 | 0.24444 | 0.26510 | 0.22513 | Y |

Notes:
I.A.M, infinite allele model; T.P.M, two phase model; S.M.M, stepwise mutational model, estimation indicate the mutation in stepwise mutational model; Y, yes; N, no; Significant values $P < 0.05$.

populations, while the parallel statement was assumed using the stepwise mutation model (S.M.M). Moreover, the shift-mode estimation of allele frequencies was perceived in all five populations, while altogether none of the mutation models were broadly illustrated consistent signals of bottleneck in engaged populations. Therefore, the current experiment could not detect any signals of bottleneck within the *P. canius* populations based on three applied models. Nevertheless, outcomes of mutational models consistently suggested the extension in populations due to absence of genetic drift and/or an invasive allele originating from different populations (*Piry, Luikart & Cornuet, 1999*).

## DISCUSSION

### Phylogeography and evolutionary history

This study has confirmed the efficiency of the COI barcode in identification of eel-tail catfish species. Barcoding fragment has been effectively sequenced mitochondrial DNA isolated from two species of family Plotosidae. Current study provided the first sequence database of *P. canius* to be submitted into barcoding data set. The first and the most common outcome of the undergone experiment could be the involvement of two common haplotype. While the most common sequence was KR086940 between populations of Selangor (*n* = 7), Sarawak (*n* = 6), and Johor (*n* = 9), the KR086939 identified as second common haplotype sharing between Johor (*n* = 6) and Negeri Sembilan (*n* = 4). Nevertheless, the most significant finding should be the occurrence of shared haplotype between Selangor, Johor (Peninsular Malaysia) populations and the Sarawak (Borneo) population. The haplotype sharing and their consequent gene flow could practically happen due to several reasons such as breeding migration, mutation, pelagic larvae, and sharing of common ancestors (*Frankham, 1996*).

Migration is a common behaviour in nearly 3% of all extant fish species (*Binder, Cooke & Hinch, 2011*). However, there is practically no record on migration and migratory behaviour of family Plotosidae, thus the first assumption of dispersal via migration and ocean current might be highly unlikely since majority of the catfish species cannot endure long distance swimming (more than 500 km) due to their body shape and structure (*Jónsson, 1982*). In addition, marine dispersal of eggs, larvae and even juveniles of *P. canius* between two separate ocean currents comprising Straits of Malacca (Selangor and Johor) and South China Sea currents (Sarawak) might also be questionable. In the Strait of Melaka, circulation of currents (particularly in surface) are due to effects of tidal patterns and wind, while the route of both surface and deep layer currents are shown to be relatively the same and toward northwest (*Rizal et al., 2010*). In Johor, however, currents are highly depend on strong winds during monsoon seasons. Indeed, if the monsoon is in its northeast route, the current streams toward South along the coastal region of Malaysia, otherwise, the current will be northward (*Mohd Akhir & Chuen, 2011*). Finally, pattern of ocean currents in western South China Sea are largely influenced by season. The circulation in South segment of western current tends to be stable and northward where after separation from coastal region, it forms a northeastern pattern in summer. In fall, however, it strongly flows toward southwest (*Fang et al., 2012*). Therefore, general patterns of aquatic circulation in Strait of Melaka, Johor maritime territory and western area of South China Sea might not strongly implies the probable distribution of grey eel-tail catfish eggs, larvae and juveniles and its consequent gene flow and genetic connectivity.

Considering all possible expectation on genetic variability and gene flow of *P. canius* in Peninsular Malaysia, the second hypothesis of sharing common ancestor due to historical geographic events may reflect the most plausible explanation. Southeast Asia is believed to have endured simultaneous glaciation and consequent deglaciation along with its associated decreasing and increasing of marine levels during the Pleistocene

period, which greatly influenced continental and oceanic configuration (*Voris, 2000*). During such variations, some regions might be preserved their stable environmental conditions that is nowadays called a refugium on which can greatly affect the gene flow and genetic variability particularly in endemic species (*Hobbs et al., 2013*). Moreover, geographical information proposed that the Pacific and Indian ocean were initially connected directly before the formation of Sundaland (nowadays submerged forming shallow ocean of most Southeast Asia with less than 100 m depth) during the Triassic up to the Pleistocene period (*Esa et al., 2008*), hence made such gene flow possible between these comparatively distant locations.

Demographic history analyses did not found any evidence of recent population expansion in all *P. canius* populations. Thus, the evolutionary history estimation (the point of expansion) could not be estimated but the *P. canius* populations were regarded as being in a stable state, possibly for a long period of time. Nevertheless, comparison of $\theta_1$ and $\theta_0$ parameters for all populations indicates a relatively slow growth rate in female populations of *P. canius* except in the Sabah population.

## Hardy-Weinberg equilibrium and genetic diversity

Overall allelic richness revealed quite lower rates using the cross amplified primers ranging from 2–8 among the sampled populations in comparison with original species (*Rourke et al., 2010*). Tan 3-28 demonstrated the highest overall allelic rate fluctuating from 1–3 among five populations of *P. canius,* while the lowest level was detected in Tan 3-27. Moreover, the Selangor population showed maximum number of alleles (9), whereas the Johor and Sarawak populations exhibited the lowest (6). Similar instance of low allelic variation have been described in *Bolbometopon muricatum* (*Priest et al., 2014*), *Schizothorax biddulphi* (*Palti et al., 2012*) and *Prosopium cylindraceum* (*Mccracken et al., 2014*). A possible reason for the occurring of low levels of allelic richness might be due to the small employed population size. *Hale, Burg & Steeves (2012)* pointed out the positive effects of sampling size between 25–30 individuals per population, however they also mentioned the necessity of 5–100 samples per collection to avoid rare uninformative alleles. Marine vertebrates are believed to present greater allele difference at their microsatellite primers comparing to freshwater populations, which is mostly consistent with their higher population evolutionary size (*Neff & Gross, 2001*). Their research later revealed that the difference in microsatellite polymorphism among classes and species could be highly dependent upon changes in life history and population biology and moderately to differences happening to microsatellite functions during natural selection. Therefore, the fewer number of allele found in *P. canius* might be due to variation in its biology and historic traits, however, the correlation of allelic richness and sample size should not be overlooked.

The average value of $H_o$ estimated in the five tested microsatellites in *P. canius* were as low as 0.2168, which showed high difference levels in comparison to standard heterozygosity in marine populations ($H_o = 0.79$) and anadromous fish species ($H_o = 0.68$) (*DeWoody & Avise, 2000*). In fact, considerable heterozygote deficiencies were observed in the engaged populations with the exception of the Tan 1-7 and Tan 3-28 loci. Similar

temporal pattern of low genetic diversity have been reported for *Pleuronectes platessa* in Island (*Hoarau et al., 2005*) and *Clarias macrocephalous* (*Na-Nakorn et al., 1999*), while in most catfish species higher levels of heterozygosity have been documented as in *Mystus nemurus* ($H_o$ = 0.44–0.57) (*Usmani et al., 2003*). Several decisive issues might influence the genetic variability of marine species through the variation of Hardy-Weinberg including migration, genetic drift, sample size, over-exploitation, effective size of population and patterns of mating (*DeWoody & Avise, 2000*). Certainly, *P. canius* should not be presumably considered as long distance migratory marine fish species due to its body structure (*Riede, 2004*). Alternatively, a possibility of genetic drift in the current study is also suspicious as it basically happens only in small effective size populations that experiencing a period of bottleneck (*DeWoody & Avise, 2000*) at which is completely invalidated in marine species studies like current research.

Small sample size of collected populations might also be measured as a major parameter in detection of low heterozygosity variation because of the failure to accurately detect the entire extant alleles of the selected populations, hence, deficiency in heterozygote identification (*Na-Nakorn et al., 1999*). Indeed, the current collection size for *P. canius* used for population genetic analysis purposes should be quite small based on *Kalinowski (2005)*; therefore, the hypothesis of deficiency in heterozygote detection due to the low level of sampled specimen could be accepted. The last cause of a low heterozygosity levels and its consequent genetic variation is non-random system of mating behaviour among populations (*Brook et al., 2002*; *Balloux, Amos & Coulson, 2004*). Estimation of HWD for the current study however, showed considerable deviation for approximately 36% of the primer/population pairs, which might be due to heterozygote deficiency effects. However, *Balloux, Amos & Coulson (2004)* highlighted that the positive correlation of inbreeding and heterozygosity needs to be examined through application of more polymorphic markers on which demonstrates greater proportion of linkage disequilibrium. Alternatively, the correlation of *Fis* values and inbreeding have been practically assessed and documented in many studies (*Balloux, Amos & Coulson, 2004*; *Abdul-Muneer, 2005*; *O'Leary et al., 2013*). The positive calculated estimations could be translated as a decrease in heterozygous levels among offspring in a population, mostly due to absence of random mating and its subsequent inbreeding. The current study showed considerable significance levels ($P < 0.05$) of *Fis* estimations. This alongside with substantial departure from HWE would indicate the damaging effect of heterozygosity deficit within the populations.

### Analysis of population structure

A remarkably high levels of genetic structure were detected among populations of *P. canius* ranging from 0.05417–0.62504, showing significantly high structuring among studied populations except differences between Johor–Selangor samples ($F_{ST}$ = 0.05417) and Selangor–Negeri Sembilan ($F_{ST}$ = 0.09806). Moreover, AMOVA statistics evidently revealed that approximately 64% of genetic variations were due to within population variations. Hence, the fairly high $F_{ST}$ rates, significant hierarchical molecular results and consequent higher genetic variances among *P. canius* populations in Peninsular Malaysia

and their relatives in Borneo, in addition to the detection of only one sharing haplotype (KR086940), would suggest the absence of contemporary gene flow among them most probably due to the geological modification, consequential rise in marine water levels and historical continental separation during the Pleistocene era (*Esa et al., 2008*; *Song, 2012*). However, exceptional cases between Selangor–Negeri Sembilan and Selangor–Johor might be inversely interpreted as occurrence of gene flow or migration regarding to fairly close distances rather than extraordinary distance between Borneo and Peninsular Malaysia. The sequential genetic diversity presented in this study could be caused by high haplotype frequencies among the five populations of *P. canius* in Malaysia, in addition to identification of unique sequences in each population (except in Johor). The present patterns of differentiation among catchments is believed to be significantly as a consequence of the Pleistocene associated historical and geological continental and sea level distraction and its subsequent isolation of lands and populations (*Esa et al., 2008*).

The calculated $F_{ST}$ values of five microsatellites in *P. canius* showed significant estimation, indicating substantial genetic structure and differentiation among the studied populations. All populations also showed significantly high assignment rates, followed by a low membership recorded for other population clusters. High rates of proper assignment might indicates strong population differentiation among the studied populations (*Paetkau et al., 2004*). Although the Sabah population demonstrated a close pairwise distance with the Selangor and Negeri Sembilan populations, the Negeri Sembilan and Selangor populations showed the lowest differentiation level (0.29711), and also the highest cluster membership in comparison with other populations. Surprisingly, the highest level of pairwise $F_{ST}$ differentiation has been estimated between the Johor population and the other four populations, in contrast to the closer geographical distance between the Johor and the Negeri Sembilan populations. Indeed, microsatellite analysis made a relatively counter-outcome in comparison with mitochondrial results, where $F_{ST}$ estimation of former populations was estimated as the lowest among the *P. canius* samples. Discrepancies between genetic differentiation detection through microsatellite loci and mitochondrial DNA is believed to be related to three factors: (1) high sensibility of mitochondrial COI gene in detection of variation (*Shaw, Arkhipkin & Al-Khairulla, 2004*), (2) weaker nuclear-based subpopulation detection (*Cano, Mäkinen & Merilä, 2008*) and (3) technical complications of microsatellite like homoplasy (*Estoup, Jarne & Cornuet, 2002*).

One of the most common practical problems, which is believed to be mostly associated with microsatellite primers (due to higher mutation rate) is well-known as homoplasy (*Balloux & Lugon-Moulin, 2002*). Homoplasy might diminish the microsatellite-based population differentiation signals. The existence of homoplasy is highly dependent on the occurrence of different identical locus copies, while such identical character is not consequent of mutual ancestor. In fact, the occurrence of homoplasy might be correlated with combination effects of high rates of mutation, and outsized population together with strong restriction in allele size (*Estoup, Jarne & Cornuet, 2002*). However, the effective number of alleles on which presented in Tables 8 and 9 showed a low level of allele size frequency, the current population size of *P. canius* used for population genetic analysis is

ostensibly quite small based on *Kalinowski (2005)* rather than being oversized. Hence, the later effective cause of homoplasy is somehow nullified in this study. Furthermore, several microsatellite based studies have pointed out the significance of S.M.M on possibility of homoplasy in different taxa (*Angers & Bernatchez, 1997*; *Culver, Menotti-Raymond & O'Brien, 2001*; *Estoup, Jarne & Cornuet, 2002*; *Anmarkrud et al., 2008*), which was invalidated by provided statistics on bottleneck analysis of recent study in Table 14.

*O'Reilly et al. (2004)* later pointed out that implications of homoplasy in identification of population structure using microsatellite loci are supposedly common in marine species. Nevertheless, further researches have been implied that implications of genetic drift and gene migration might have considerably greater effects on population differentiation analysis in comparison with homoplasy (*Estoup, Jarne & Cornuet, 2002*). Basically, marine vertebrates supposed to have the higher population effective size ($N_e$) comparing to freshwater species (*Hauser & Carvalho, 2008*). Besides, genetic drift and effective size are believed to be greatly correlated, hence it is highly probable that neighbouring geographical populations demonstrate the imperceptible population differentiation and structures especially using neutral primers like microsatellites (*Larmuseau et al., 2010*).

## CONCLUSION

The current genetic characterization of *P. canius* provided some basic results on their phylogeny and population structure. The phylogenetic and phylogeographic analysis of *P. canius* noticeably constructed accurate clusters in the five population of Malaysia, demonstrating the capability of the chosen mitochondrial COI barcoding gene to potentially assign the genus *Plotosus* into their biological taxa. Indeed, COI analysis resolved the phylogenetic relationships between *P. lineatus* and *P. canius* populations, supporting their taxonomic status as different species. A low mitochondrial differentiation was found among *P. canius* populations with some indication of endemism (haplotype restricted only to a particular population) as observed in the Sabah population. Nevertheless, the COI gene revealed sufficient informative interpretation of relationships among the five populations, supporting by reasonable bootstrapping values (> 85%). The sharing of haplotypes between a few samples from Peninsular Malaysia and their Sarawak counterpart of Borneo showed the sensitivity of the COI marker to infer the biogeographical history of *P. canius* and potentially other marine taxa in the region.

Microsatellites analyses on the population structure of *P. canius* showed slightly different patterns of structuring in comparison with the COI findings. Nevertheless, both markers detected higher level of among population differentiations than within population variability. In addition, four main clusters or genetic stocks of *P. canius* were identified using the cross species amplification study of *T. tandanus* microsatellites.

Finally, the results from this study has provided valuable understandings on the genetic characterization, molecular phylogeny, evolutionary kinship, and population structuring of *P. canius*, in particular, and the genus *Plotosus*, in general. However, further studies must be conducted using more geographical and sampling sites, larger population sizes

per site. Furthermore, designing species specific hypervariable nuclear markers such as microsatellite for *P. canius* must be considered in order to accurately analyze the population structure and genetic diversity of *P. canius* before implementation of advanced fisheries and conservation strategies.

### Funding
The authors received no funding for this work.

### Competing Interests
The authors declare that they have no competing interests.

### Author Contributions
- Nima Khalili Samani conceived and designed the experiments, performed the experiments, analyzed the data, wrote the paper, prepared figures and/or tables.
- Yuzine Esa conceived and designed the experiments, analyzed the data, contributed reagents/materials/analysis tools, reviewed drafts of the paper.
- S.M. Nurul Amin contributed reagents/materials/analysis tools, reviewed drafts of the paper.
- Natrah Fatin Mohd Ikhsan reviewed drafts of the paper.

### Animal Ethics
The following information was supplied relating to ethical approvals (i.e., approving body and any reference numbers):

Fish species that has employed in this study is not under endangered list. Besides, there is no institute in Malaysia to issue any kind of approval letter for catchment or approval of research on involved species.

### Data Deposition
Relevant accession numbers, raw data and other associated details will be publicly accessible on GenBank and BOLDSYSTEM after publication of manuscript. Accession numbers are listed below:

BOLDSYSTEM Index GenBank Accession

NUPM017-14 KP258659, NUPM016-14 KP258657, NUPM015-14 KP258658, NUPM001-14 KP258648, NUPM002-14 KP258651, NUPM006-14 KP258655, NUPM023-15 KR086935, NUPM003-14 KP258650, NUPM004-14 KP258649, NUPM005-14 KP258656, NUPM022-15 KR086936, NUPM007-14 KP258654, NUPM008-14 KP258653, NUPM009-14 KP258652, NUPM020-15 KR086938, NUPM021-15 KR086937, NUPM010-14 KP221601, NUPM011-14 KP221602, NUPM012-14 KP221603, NUPM013-14 KP221604, NUPM014-14 KP221605, NUPM018-15 KR086940, NUPM019-15 KR086939.

## Supplemental Information

Supplemental information for this article can be found online at http://dx.doi.org/ 10.7717/peerj.1930#supplemental-information.

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
