# Peer review of "Phylogenetics and population genetics of Plotosus canius (Siluriformes: Plotosidae) from Malaysian coastal waters"

_PeerJ, doi:10.7717/peerj.1930_

## Round 0.1 · original submission · Major Revisions

Please consider all the suggestions in the revised manuscript.

Reviewer 1 ·

Basic reporting

Interesting view on molecular variability of Plotosus canius in Malaysia. However, in my opinion, more fundamental flaws in the MS that need to be seriously dealt with, before it can be considered for publication (see my comments to the authors)

Experimental design

See my comments to the authors

Validity of the findings

See my comments to the authors

Additional comments

Title
- Suggest to revise as ' Phylogenetics and Population Genetics of Plotosus canius (Siluriformes: Plotosidae) from Malaysian coastal waters

Abstract
- It is not correct to interprete the values of haplotype diversity and nucleotide diversity for unlikelihood of mutation effects on gene flow.
- Line 50: I do not understand the logical basis the authors suggest the utilization of ‘more COI genes’?

Introduction
--The introduction is too lengthy. The reader would often appreciate a clear and concise introduction in relation to the background of study.
--Line 103-106: what is the relevance of phylogenetics?

Materials and Methods

-- The heading ‘Statistical analysis of Mitochondrial DNA’ is not applicable in the context of PCR amplification, purification and sequencing.
-- A heading ‘data analyses’ should be created for mitochondrial and microsatellite analyses.
-- Using Neighbour joining to infer tree and only NJ, is surprising! Especially since the authors used the program MEGA. The author should also, at least, make some ML or Bayesian trees. I concede that trees may not be changed but nonetheless NJ alone is below-par.
Results
-- I would suggest naming the haplotypes as H1, H2, H3 etc...instead of accession numbers.
--Figure 2, poor figure quality....and in fact, this is a table..
--Figure 3 legend, use of ‘haplotypes’ for three species is not correct.
-- Line 311-312: which data allow to conclude ‘decrease in gene flow’ ?
-- I cannot found KR086940 as the most common haplotype in the entire populations’ neither in the tables nor in the figures. In this contetx a bit strange that the authors did not provide the frequency of haplotype distribution for the studied taxa.


Discussion
-- shared haplotype is a common observation for same species. In contrast, absence of ancestral haplotype between two or more geographically distinct areas is an interesting finding.
-- how the authors infer gene flow based on the sharing of haplotype?
-- ‘sharing common ancestor due to historical geographic events’, how? It is strange that the authors relate their findings with historical events (Pleistocene glacial), without considering the use of their sequence data for a demograhic analysis.
-- the authors claim that their microsatellite data might be influenced by homoplasy. If this is the case, I would rate this data as invalid due to the poor phylogenetic signal. I suggest to perform a test, to rule out the possibility of homoplasy. At least this information should be mentioned in Materials and Methods.

·

Basic reporting

The manuscript is written very well and all the data were well presented. The concern is how did the species identification done and is it clear distinction of morphology when the collection was done. The reasons I ask is there a possibility of misidentification maybe there could be subspecies, hybrids, cyptic speciation. I hope the all the authors took some steps to be vigilant on the identification process.

Experimental design

yes the experimental design was sound> However the collection was done from local fish markets and how sure is the collection method coming from that geographical location. Did you buy from the fisherman directly from the sea or from the markets where thefish are frozen brought from different locations? This should be important as the practice here is fish from local market does not originated locally.

Validity of the findings

The work is statistically sound but the use of 4 microsatellite loci is not enough to demonstrate the genetic status and moreover the it is cross amplified microsatellite markers. therefore can you provide the PIC value of microsatellite markers so that the results will be more reliable.No of effective alleles is very low and it does not make sense if you look at nucleotide diversity which is higher. I strongly suggest that markers are not strong enough to show differentiation .

Additional comments

The COI data is good but does not reflect the geographical sites but I need to be convinced more on the microsatellite data. Please provide the PIC information of microsatellites on add a another neutral marker if possible. I am not convinced with the data as you said Sabah populations are close to Negeri Sembilan and Selangor. Johor is furthest..It must be sampel collection is not in accordance to geographical conditions...Please convince me that you are sure the samples are collected from that geographical location.

---

## Round 0.2 · Minor Revisions

Your manuscript has been re-reviewed by the 2 original reviews. As you can see, reviewer 1 still requests some additional edits.

Reviewer 1 ·

Basic reporting

This revision has some improvements over the original manuscript. However, the explanation of the data analyses remains unclear and was reported in a very confusing way

Experimental design

No Comments

Validity of the findings

No Comments

Additional comments

1.There is only one COI gene, what is ‘more COI genes’ or ‘extra COI gene’ means for? Perhaps authors would like to propose‘ other COI regions’?, if this is the case, I would suggest other genes to be tested, instead of focusing COI gene.

2. Again, this is not a figure!!! The authors did not consider alternate ways such as microsoft word or excel to generate the table!

3.Figures 3 and 4 can be combined into one.

4.Table 2- reported in a very confusing way, the frequencies of haplotype should be expressed in %

5. The authors failed to provide evidence for historical events based on their own dataset. Although they provided some references but the information are not applicable to their studied taxa. I would suggest to remove all historical and demographic content in the ms.

6.Author responses to reviewer’s comments:
"Furthermore, the questioned issue on demographic analysis was already presented in manuscript by presenting a straightforward geographical analysis using the Minimum Spanning Network (Figure.5) mentioning that due to occurrence of mix haplotypes with other clades. This basically indicating that no accurate geographical genetic structure have been certainly detected in the studied populations of P.canius in Malaysia."

--I think the authors might have misunderstood the common term for ‘demographic analysis’. The response from the authors, towards the Minimum Spanning Network results, is just the indicative for population genetic structure, not ‘demographic’ or specifically ‘demographic history’.

·

Basic reporting

The authors have answered the rebuttal and I am convinced

Experimental design

Experimental design is sound

Validity of the findings

The findings is valid with the use of mitochondrial and microsatellites

---

## Round 0.3 · accepted · Accept

Thank you for improving your manuscript.